

# Critical colored-RVB states in the frustrated quantum Heisenberg model on the square lattice

**Didier Poilblanc\*, Matthieu Mambrini and Sylvain Capponi**

Laboratoire de Physique Théorique, Université de Toulouse, CNRS, UPS, France

\* didier.poilblanc@gmail.com

## Abstract

We consider a family of SU(2)-symmetric Projected Entangled Paired States (PEPS) on the square lattice, defining colored-Resonating Valence Bond (RVB) states, to describe the quantum disordered phase of the $J_1 - J_2$ frustrated Heisenberg model. For $J_2/J_1 \sim 0.55$ we show the emergence of critical (algebraic) dimer-dimer correlations – typical of Rokhsar-Kivelson (RK) points of quantum dimer models on bipartite lattices – while, simultaneously, the spin-spin correlation length remains short. Our findings are consistent with a spin liquid or a weak Valence Bond Crystal in the neighborhood of an RK point.



# 1 Introduction: RVB and the frustrated Heisenberg model

Resonant valence bond (RVB) states were first proposed by Anderson [1] to describe a possible spin liquid ground state of the $S = 1/2$ antiferromagnetic Heisenberg model on the triangular lattice. Later on, it was also introduced as the parent Mott state of high-$T_c$ superconductors [2]. Soon after, along the same spirit, the Rokhsar Kivelson (RK) wavefunction [3] was defined as equal weight superposition of nearest neighbor (NN) dimer coverings, avoiding an explicit reference to the (hidden) spin degrees of freedom. It was shown that the RK wavefunction is a *critical* dimer liquid state [4] on the square lattice, in contrast to the case of non-bipartite kagome and triangular lattices [5–8] on which a gapped ($\mathbb{Z}_2$) dimer liquid state is realized instead. Similarly, the NN RVB state, defined as an equal weight superposition of (non-orthogonal) NN singlet bond (also dubbed "dimer") coverings, was shown to be also critical on the square lattice [9,10] while several numerical work [11–14] have demonstrated that their analogs on the kagome and triangular lattices are $\mathbb{Z}_2$ spin liquid states. Note that the (dimer) critical RK point is commonly unstable – ie towards dimerized phases [3,15] or gapped dimer liquid phases [16] – upon slightly varying the model parameters. In fact, generically the RK point appears to be a multi-critical point with all sorts of nearby phases in which the critical correlations present at the RK point could be correct over a substantial intermediate range of energy and length scales. SU(2)-invariant spin models have also been engineered [17,18] to mimic quantum dimer physics on the square lattice, with (critical) RVB ground state and Valence Bond Crystal (VBC) phases (spontaneously breaking translation symmetry), reflecting also the multi-critical nature of the RK point in SU(2)-symmetric systems.

Spin liquid behaviors are expected in two-dimensional (2D) frustrated quantum magnets where magnetic frustration prohibits magnetic ordering at zero temperature. Strong magnetic frustration is realized in the square lattice $J_1 - J_2$ spin-1/2 Heisenberg model defined by summing over a 2D grid of lattice points $(i, j)$,

$$H = \sum_{i,j} \left[ J_1(\mathbf{S}_{(i,j)} \cdot \mathbf{S}_{(i+1,j)} + \mathbf{S}_{(i,j)} \cdot \mathbf{S}_{(i,j+1)}) + J_2(\mathbf{S}_{(i,j)} \cdot \mathbf{S}_{(i+1,j+1)} + \mathbf{S}_{(i+1,j)} \cdot \mathbf{S}_{(i,j+1)}) \right] \quad (1)$$

and including both NN and next nearest neighbor (NNN) antiferromagnetic couplings $J_1$ (set to 1) and $J_2$, respectively. A paramagnetic quantum disordered (QD) region was suggested by early Lanczos Exact Diagonalizations (ED) extrapolations (including up to $N = 36$ spins) in the range $J_2 \in [0.34, 0.68]$ [19], and similar results were announced later using ED up to $N = 40$ [20]. However, until now, no agreement has been reached between several numerical approaches on the nature of the QD region – with proposals of VBC [19, 21–24], (topological) gapped [25,26] or gapless [24,27–29] spin liquids. Interestingly, density matrix renormalization group (DMRG) approaches [30] with explicit implementation of SU(2) spin rotation symmetry [24] suggest that the QD region splits into a (critical) spin liquid phase (for $0.44 < J_2 < 0.5$) and a plaquette VBC phase (for $0.5 < J_2 < 0.61$). Recently, DMRG simulations of Wang and Sandvik using level spectroscopy [31] also indicate that the QD is formed by a gapless spin liquid phase (for $0.46 < J_2 < 0.52$) and a VBC (for $0.52 < J_2 < 0.62$). In contrast, other recent computations using U(1)-symmetric (infinite size) Projected Entangled Pair States (PEPS) [32] suggest a columnar VBC (for $0.53 < J_2 < 0.61$) separated from the conventional Néel phase by a deconfined critical point [33], in qualitative agreement with a previous finite size PEPS computation [34].

Despite such recent progress, the exact nature of the QD phase remains still unclear. In this paper we aim to investigate further the QD phase in the region around $J_2 = 0.55$ introducing simple PEPS Ansätze which are specially designed to describe SU(2)-invariant states with full space group symmetry. In Sec. 2 we quickly review the iPEPS method used, further details being provided in Appendix A. Results on variational energy and correlation functions are analyzed in Sec. 3 and complementary ED results are provided in Appendix B, strongly suggesting

that RK physics with long-range dimer correlations emerges. Finally, further discussions and conclusions are given in the last section 4.

## 2  Numerical implementation

### 2.1  iPEPS method

Tensor networks [35–38] have recently emerged as a state-of-the-art numerical tool to tackle correlated lattice models. Among them, 2D PEPS [39,40] are variational Ansätze constructed from local site tensors carrying the physical degrees of freedom (of dimension 2 for spin-$\frac{1}{2}$ systems) and $z$ "virtual" bonds ($z$ is the lattice coordination number, $z = 4$ for the square lattice) of arbitrary dimension $D$ (see Appendix (A)). Interestingly, local (gauge) or global (physical) symmetries can be implemented in PEPS [41–48]. In the infinite-PEPS (iPEPS) method [49], one works directly in the thermodynamic limit by approximating the (infinite) space around a small $M$-site cluster by an effective "environment" (here $M = 2 \times 2 = 4$). One of the most accurate computation of the environment is based on a real-space Renormalization Group (RG) scheme involving Corner Transfer Matrices (CTM) [50–53], the so-called CTMRG algorithm. Unrestricted energy minimization can be performed [54,55] using a simple update [56,57] or a full update [58] of the environment. Recently, a new variational optimization scheme using a Conjugate Gradient (CG) algorithm has been tested on the non-frustrated [59,60] and on the above spin-1/2 $J_1 - J_2$ Heisenberg model [61,62].

### 2.2  Colored-RVB states

We wish here to refine and extend the previous iPEPS study of Ref. [62] dealing with the spin-1/2 frustrated $J_1 - J_2$ Heisenberg model on the square lattice. While Ref. [62] focused mainly on $J_2 = 0.5$ – pointing towards a gapless spin liquid – we focus here on slightly larger $J_2 \sim 0.55$ where, we shall argue, a new behavior occurs. For this purpose, we shall consider the same families of translationally invariant fully symmetric PEPS involving a linear combination of a finite number $\mathcal{D}$ of single site tensors,

$$a = \sum_{\alpha=1}^{\mathcal{D}} c_\alpha t_\alpha. \tag{2}$$

The tensors $t_\alpha$, obtained from a complete classification of symmetric site tensors on the square lattice [48], are fully invariant under SU(2) spin rotations and under all operations of the $C_{4v}$ point group (90-degree rotations and reflections). These local symmetry properties of the site tensors guarantee that the PEPS itself is a fully symmetric wavefunction under all the global symmetry operations leaving the Hamiltonian invariant. The bond virtual space of dimension $D = 2\mathcal{N} + 1$ is of the form

$$V = \overbrace{\frac{1}{2} \oplus \cdots \oplus \frac{1}{2}}^{\mathcal{N} \text{ times}} \oplus 0, \tag{3}$$

corresponding to $\mathcal{N}$ possible "colors" of spin-$\frac{1}{2}$ and a spin-0 (singlet). In the following we shall consider the three PEPS families associated to one, two and three colors of the spin-1/2 degree of freedom i.e. namely to $V = \frac{1}{2} \oplus 0$ ($\mathcal{N} = 1, D = 3$), $V = \frac{1}{2} \oplus \frac{1}{2} \oplus 0$ ($\mathcal{N} = 2, D = 5$), and $V = \frac{1}{2} \oplus \frac{1}{2} \oplus \frac{1}{2} \oplus 0$ ($\mathcal{N} = 3, D = 7$). Each of these PEPS family is spanned by a small number $\mathcal{D}$ of linearly independent tensors, $\mathcal{D} = 2$, 10 and 30 respectively, given in Refs. [48,62].

It is important to notice that these three PEPS families are not separated from each other but rather embedded into one another. The smallest one (spanned by 2 independent $D = 3$ site

Figure 1: $\mathcal{N}$-color RVB manifolds ($\mathcal{N} = 1, 2, 3$) spanned by onsite PEPS tensors involving, on their four virtual bonds, one, two and three colors (here blue, red and green) of spin-1/2 virtual degrees of freedom. The spin-0 degrees of freedom (site physical variables) are shown as dashed virtual legs (grey bullets) and the number of linearly independent (point group-symmetric) tensors of each kind is shown in parenthesis. All tensors contain either one or three spin-0 legs. The inner ensembles correspond to three (identical) copies of the $V = \frac{1}{2} \oplus 0$ PEPS manifold (each spanned by two site tensors). Each of the three copies of the $\mathcal{N} = 2$ PEPS family is spanned by $2 \times 2 = 4$ single-color tensors and, simultaneously, by a set of six 2-color tensors. The $D = 7$ 3-color RVB subspace is spanned by all the 30 tensors of the picture.

tensors) can be viewed as a manifold of generalized RVB states which include, when expanded in terms of valence bond (VB) configurations, singlet bonds extending beyond NN sites (in contrast to the original NN RVB state [1]). Its corresponding phase diagram contains a RK dimer liquid phase and a (topological) spin liquid phase [63]. The $\mathcal{N} \geq 2$ PEPS family can be viewed as $\mathcal{N}$-color (generalized) RVB states where singlet valence bonds (VB) carry now a color index, ranging from 1 to $\mathcal{N}$, and where the VB amplitudes depend on the coloring

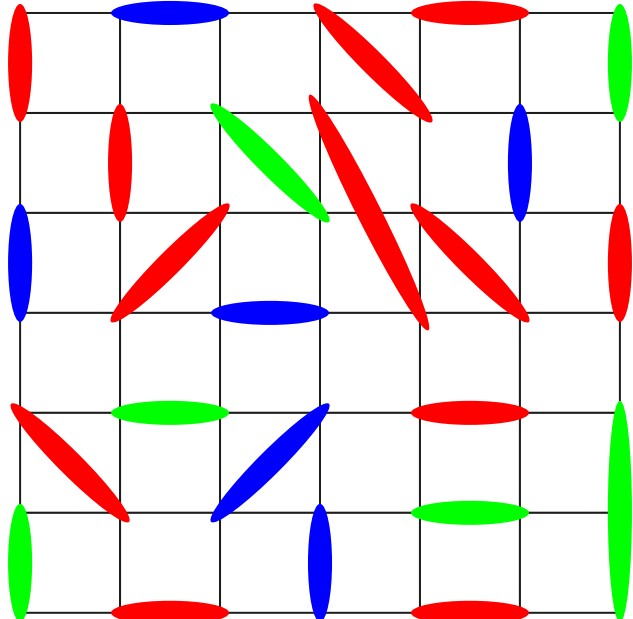

Figure 2: A typical VB covering of the 3-color RVB state whose amplitude depends *both* on the singlet covering and on the coloring pattern.

pattern. In that new language, it becomes obvious that the $\mathcal{N} = 2$ PEPS family includes two (disjoint) copies of the manifold of single-color RVB states. Similarly, our largest $\mathcal{N} = 3$ PEPS family – defining the manifold of 3-color RVB states – contains three (disjoint) sub-manifolds of single-color RVB states (of different colors) and three (disjoint) sub-manifolds of 2-color RVB states (with different pairs of colors). These features are summarized in Fig. 1.

All site tensors contain either one or three spin-0 virtual legs (leading to a $\mathbb{Z}_2$ gauge symmetry [63] associated to the odd parity of this number of legs). Note that, if one restricts to only the subset of tensors with a single spin-1/2 leg and three spin-0 legs, the corresponding PEPS is the usual NN RVB state (the VB amplitudes do not depend on the coloring pattern in that case). Longer range singlets are created by "teleportation" [28, 63] introduced by any of the site tensors containing three spin-1/2 (of any color) and one spin-0 on the virtual legs. Hence, the most general 3-color ($D = 7$) RVB Ansatz can be viewed as a resonant state of colored VB coverings of the type drawn in Fig. 2. The VB amplitude depends both on the VB covering and on the coloring pattern in a complex way set up by the tensor coefficients $c_\alpha$ entering Eq. 2.

## 2.3 CTMRG algorithm

For a given PEPS realization (i.e. defined by a particular set $\{c_\alpha\}$ of coefficients in Eq. 2) the corresponding energy $E[\{c_\alpha\}]$ (in the thermodynamic limit) is computed by a CTMRG method which takes advantage of the point group symmetry of the lattice (see Appendix (A) for details). Note that, although the site tensor is fully SU(2)-invariant, the CTMRG procedure of Ref. [62] (used to contract the infinite tensor network outside a $2 \times 2$ active region) was generically converging to a fixed-point environment exhibiting a small finite staggered magnetization i.e. spontaneously breaking SU(2) spin-rotation symmetry, at least for $J_2 = 0.5$ which was extensively studied. Although, this effect is spurious (the data are consistent with a vanishing staggered magnetization in the limit of infinite environment dimension, $\chi \to \infty$), it complicates the analysis of the data. Hence, we have improved the CTMRG procedure in

order to keep the full SU(2) symmetry at all stage and for all $\chi$ (despite numerical rounding errors) so that the fixed-point solution of the CTMRG indeed corresponds to a fully symmetric QD state.

### 2.4 Optimization over the tensor parameters

In summary, the CTMRG algorithm enables to compute the energy density $E[\{c_\alpha\}]$, in the thermodynamic limit, for a given choice of (i) the bond dimension $D$, (ii) the associated $\mathcal{D}$ tensor coefficients $c_\alpha$ and (iii) the environment dimension $\chi$. We use a brute force (Conjugate Gradient) optimization upon the set of coefficients $c_\alpha$ to obtain the absolute minimum of the variational energy at given $D$ and $\chi$. This requires to numerically compute each component of the local gradient vector $\vec{\mathcal{G}}$ of the energy $E[\{c_\alpha\}]$ by finite differentiation,

$$\mathcal{G}_\beta \equiv \frac{\partial E}{\partial c_\beta} \simeq \frac{E\left[\{c_\alpha\}_\beta\right] - E[\{c_\alpha\}]}{\delta}, \tag{4}$$

where, in the set of parameter $\{c_\alpha\}_\beta$, only $c_\beta$ has been incremented to $c_\beta + \delta$, $\delta/c_\beta$ corresponding typically to a relative change of less than 1%. Note that in the calculation of $E\left[\{c_\alpha\}_\beta\right]$ it is crucial to take into account the change of the environment by computing the new CTMRG fixed point. Interestingly, we observe that the color-exchange symmetry is broken in the optimal PEPS. The optimization is performed for each choice of $\mathcal{N} = 1, 2, 3$ and up to some maximum value of $\chi$, $\chi_{\text{opt}}(D)$.

Thanks to the refinements of the iPEPS technique mentioned above, we have obtained accurate results for $J_2 \sim 0.55$ detailed below.

## 3 Results

### 3.1 Energetics

The variational energies (per site) at $J_2 = 0.55$ are shown in Fig. 3(a) as a function of the inverse of the environment dimension $\chi$. Note that the local tensors are fully optimized up to a maximum bond dimension $\chi_{\text{opt}} = 12D^2 = 108$ for $D = 3$, $\chi_{\text{opt}} = 4D^2 = 100$ for $D = 5$ and $\chi_{\text{opt}} = 2D^2 = 98$ for $D = 7$. Then, energies are computed for larger environment dimensions $\chi > \chi_{\text{opt}}$ using the fixed optimized tensors obtained at $\chi = \chi_{\text{opt}}$. Linear fits can be performed in $1/\chi$ to provide the $\chi \to \infty$ true variational energies. Note that our energy $-0.4842$ for $D = 7$ is quite close to the value $-0.4856(1)$ obtained using $D = 9$ finite PEPS cluster update [34] and $D = 8$ finite PEPS Variational Monte Carlo (VMC) [64, 65]. Moreover, the $D \to \infty$ extrapolation shown in Fig. 3(b), using either a Taylor series or a power-law in $1/D$, gives $-0.4894(5)$, significantly below the DMRG [24] and the VMC [29] estimates (reported in Fig. 3(b) for convenience). Alternatively, a power-law extrapolation (almost linear) w.r.t. $1/\mathcal{N}$ gives a slightly lower energy $-0.4909$. This gives us some confidence that the series of colored-RVB states, as defined by the PEPS construction, provides a faithful representation of the low-energy physics of the $J_1 - J_2$ model at frustration $J_2 \sim 0.55$.

### 3.2 Dimer-dimer correlations

After optimizing our symmetric PEPS Ansatz w.r.t. the coefficients of the site tensor, correlation functions can be computed using arbitrarily long (let's say horizontal) one dimensional strips bounded by environment tensors [62] (which depend on $\chi$) as depicted in Fig.7 (e). Let us

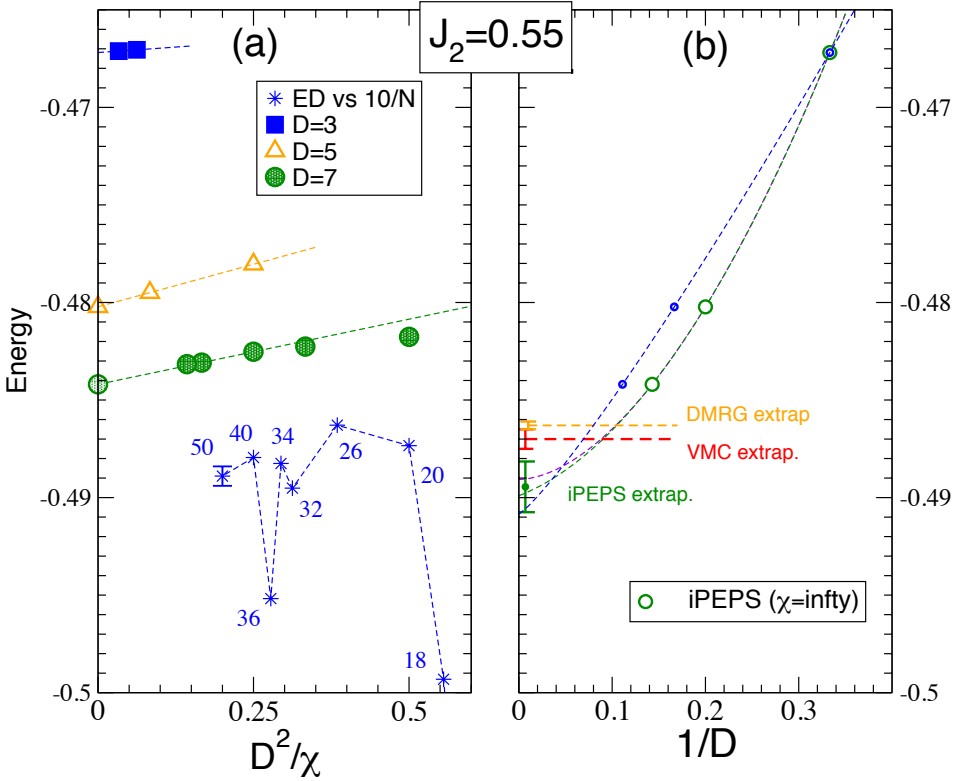

Figure 3: (a) Variational iPEPS energies plotted vs $D^2/\chi$ for $V = \frac{1}{2} \oplus 0$ ($D = 3$), $V = \frac{1}{2} \oplus \frac{1}{2} \oplus 0$ ($D = 5$) and $V = \frac{1}{2} \oplus \frac{1}{2} \oplus \frac{1}{2} \oplus 0$ ($D = 7$). Dashed lines are linear fits. (b) $\chi \to \infty$ extrapolated iPEPS energy plotted vs $1/D$. Polynomial and power-law fits give very similar $D \to \infty$ extrapolations. The same data are also plotted vs $\frac{1}{\mathcal{N}}$ ($\times \frac{1}{3}$) using smaller blue dots. An (almost linear) power-law extrapolation w.r.t. $1/\mathcal{N}$ gives a slightly lower energy. Comparison with finite size Lanczos ED of N-site square-shaped tori (plotted vs $1/N$) in (a) and with DMRG [24] and VMC extrapolations [29] in (b) are shown. Note that error bars are included in the 50-site ED energy data obtained by Lanczos step extrapolation after 30 steps.

first define the connected dimer-dimer correlations,

$$C_{\mathrm{d}}(r) = \left\langle D_{\mathbf{x}} D_{\mathbf{x}+r\mathbf{e}_x} \right\rangle - \left\langle D_{\mathbf{x}} \right\rangle \left\langle D_{\mathbf{x}+r\mathbf{e}_x} \right\rangle, \tag{5}$$

where $\mathbf{x} = (i, j)$ is some arbitrary lattice site, dimer operators $D_{\mathbf{x}} = \mathbf{S}_{\mathbf{x}} \cdot \mathbf{S}_{\mathbf{x}+\mathbf{e}_x}$ are oriented along the horizontal $\mathbf{e}_x = (1, 0)$ direction, and the expectation values are taken in the optimized PEPS. Note that, the PEPS being invariant by lattice translation, the dimer density $\left\langle D_{\mathbf{x}} \right\rangle$ does not depend in fact on the position $\mathbf{x}$. Also, although we are using a strip geometry, the local tensor (and the corresponding environment tensor $T$) has been optimized for the fully rotationally invariant (infinite) lattice.

The dimer correlations are plotted in Fig. 4(a) for $D = 7$ and $J_2 = 0.55$ and several values of $\chi$, in semi-log scale to reveal the long-distance exponential decay. From a linear fit, one can extract the corresponding dimer correlation length $\xi_d(\chi)$. The latter is plotted in Fig. 5(a) as a function of $\chi$ and, in Fig. 5(b), versus $\chi/D^2$ which seems to be the natural rescaled variable to compare the behaviors of the 3 different families $D = 3, 5, 7$. For all cases, we observe a clear linear dependence with $\chi$,

$$\xi_d(\chi) \sim a_D \frac{\chi}{D^2} + b_D, \tag{6}$$

suggesting a divergence of the correlation length, or at least saturation to a very large value beyond reach. It is interesting to notice that, once plotted in terms of the rescaled variable $\chi/D^2$, the slope $a_D$ of the linear increase is quite similar for $D = 5$ and $D = 7$, suggesting a robust feature of the correlations. In Fig. 5(c) we compare the $D = 7$ correlation length at different $J_2$ values, showing a more pronounced increase at $J_2 = 0.55$.

Whenever the correlation length $\xi_d(\chi)$ diverges (or becomes very large), one expect to see power-law behaviors in the correlation functions,

$$C_{\mathrm{d}}(r) \sim r^{-\alpha_{\mathrm{d}}}, \tag{7}$$

in the range of distance $1 < r < \xi_d$. The exponent can be written as $\alpha_{\mathrm{d}} = 1 + \eta_{\mathrm{d}}$ where $\eta_{\mathrm{d}}$ defined e.g. in Ref. [66] is the anomalous dimension. Since the correlation length remains moderate for $D = 7$, we have (i) first extrapolated the data in the $\chi \to \infty$ limit in Fig. 4(b) (using a power-law fit in $1/\chi$) for a few distances $r$ and (ii) fitted these extrapolated values to obtain the exponent $\alpha_d \sim 1.08$ via a power-law fit in Fig. 4(a). The smallness of the anomalous exponent $\eta_d \simeq 0.08$ reveals very long-range dimer correlations at $J_2 = 0.55$, in contrast to $J_2 = 0.5$ studied in Ref. [62]. Interestingly, quite similar behaviors are found for $D = 3, 5$ and 7 as shown in Fig. 4(c) where the dimer correlations are plotted, using a log-log scale, as a function of the rescaled distance $\tilde{r} = r/\xi_d$, for the largest attainable environment dimension $\chi$. Linear fits for $\tilde{r} < 1$ provide similar values for the exponent $\alpha_d$, between 1.15 and 1.25, in agreement (within error bars) with the previous analysis. It is also interesting to notice that these values are quite close to the value $\alpha_{\mathrm{d}} \simeq 1.16$ reported for the NN RVB state [9] and agree with recent DMRG simulations [24].

### 3.3 Spin-spin correlations

Finally, we have computed the spin-spin correlations (e.g. along the $\mathbf{e}_x$ horizontal direction),

$$C_{\mathrm{s}}(r) = \left\langle \mathbf{S_i} \cdot \mathbf{S_{i+r\mathbf{e}_x}} \right\rangle, \tag{8}$$

using the same strip geometry of an (infinite) chain of sites bounded by environment tensors $\mathcal{T}$ on the edges (Fig.7 (d)). In the original RVB picture [1] spins are correlated only at short distance via NN singlet pairing. Our results obtained for $J_2 = 0.55$, $D = 7$ and several $\chi$ values up to $\chi = 9D^2 = 441$ are shown in Fig. 6. Linear fits of the long-distance correlations (plotted in log-log scales) enable to estimate accurately the spin correlation length. The inset shows that the latter remains quite small, typically less than 2 lattice spacings, even for the largest $\chi$ at hand. The same is also true for $D = 3$ and $D = 5$. However, as shown in Fig. 4(a), the spin correlations are much stronger than the dimer correlations at short distance. This is consistent with the RVB picture where strong (resonating) singlet bonds are formed between NN sites.

## 4 Conclusion and outlook

The main findings of this iPEPS study at $J_2 = 0.55$ and $J_2 = 0.575$ are the following: i) a simple SU(2)-symmetric PEPS based on a single site tensor provides a very good variational energy; ii) its dimer correlations exhibit slow algebraic decay up to long distance; iii) its spin correlations are short range. Properties ii) and iii) are characteristic of RK physics found e.g. in the NN RVB spin liquid on any bipartite lattice.

The critical RK point is known to be unstable to small Hamiltonian perturbations [7,15,16] breaking the lattice bipartiteness. However, investigation of classical dimer models at finite temperature [8] indicates that criticality and nonbipartiteness are compatible. In fact, our $D = 3$ PEPS, which realizes exactly an extended RVB state with (inter-sublattice) longer-range

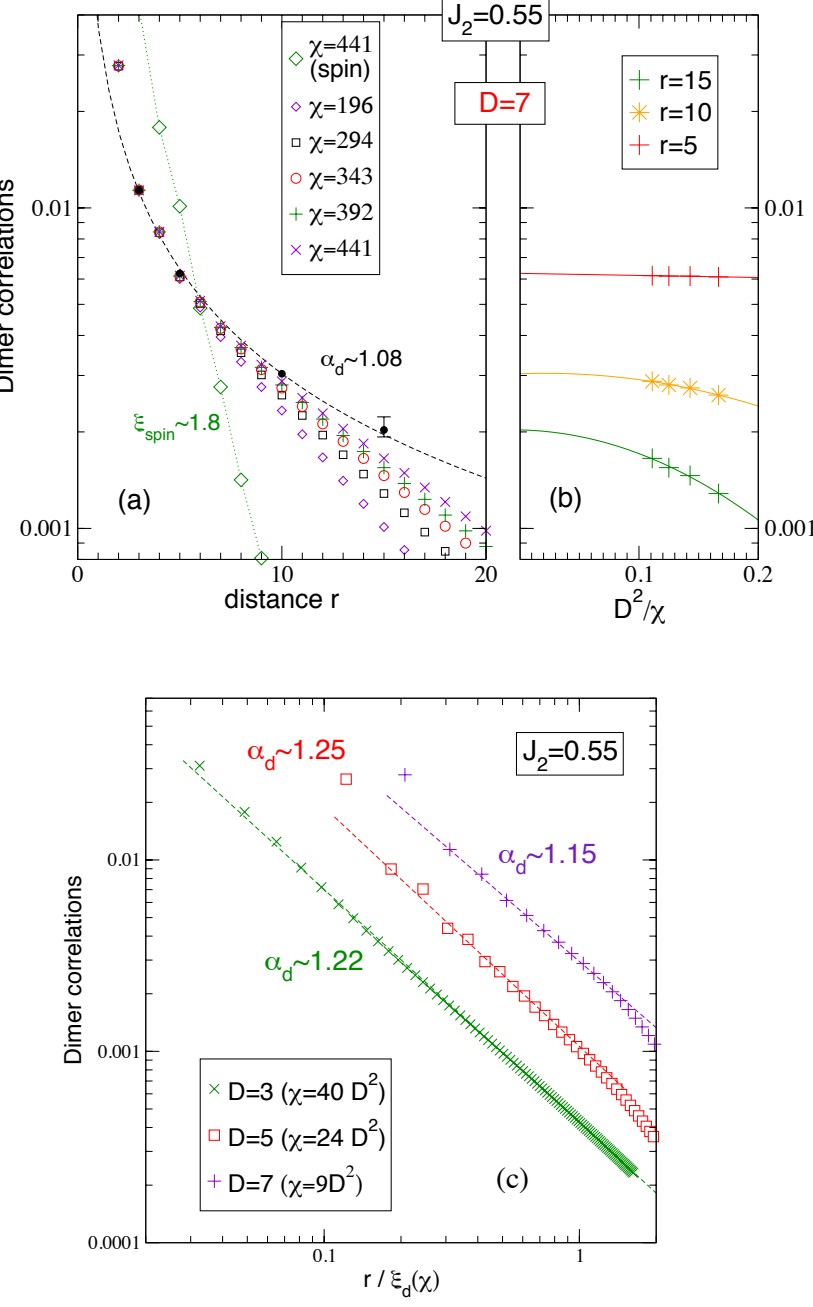

Figure 4: (a) Dimer-dimer correlations vs distance (in semi-log scale) for $J_2 = 0.55$, $D = 7$ and several environment dimension $\chi$. Spin-spin correlations at the largest $\chi$ value are also shown for comparison. (b) $\chi \to \infty$ extrapolation of the correlations at fixed distances using power-law fits in $1/\chi$. The extrapolated values are reported in (a) as black bullets fitted as a power law (dashed line). (c) Dimer-dimer correlations plotted in log-log scale as a function of the renormalized distance $r/\xi_d(\chi)$. For the values of $\chi$ used here, the dimer correlation length was found to be $\xi_d \simeq 61.5, 16.4$ and $9.65$, for $D = 3, 5$ and $7$ respectively. From the linear fits one obtains the exponent $\alpha_d$ of the power laws.

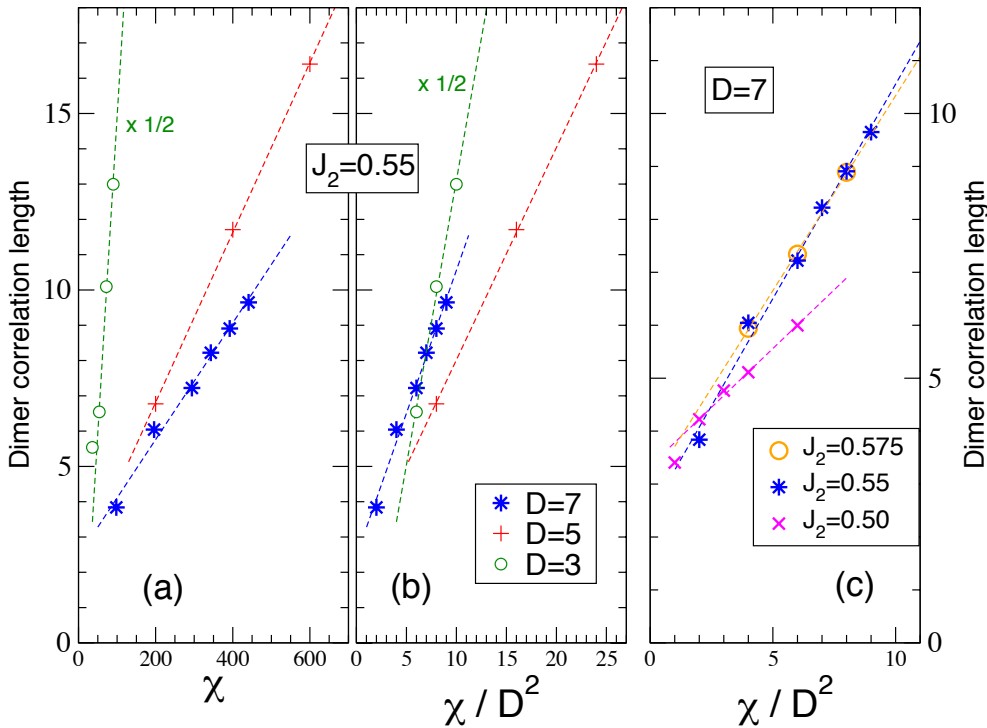

Figure 5: Dimer correlation length $\xi_d$ at $J_2 = 0.55$ plotted vs environment dimension $\chi$ (a) or vs $\chi/D^2$ (b), for $D = 3$, $D = 5$ and $D = 7$. The $D = 3$ data are multiplied by a factor $1/2$ to fit the vertical scale. (c) Comparison of $\xi_d$ vs $\chi/D^2$ for different $J_2$ values and fixed $D = 7$. Data for $J_2 = 0.5$ are taken from Ref. [62].

singlets, is known to possess an extended RK dimer liquid phase [63]. Also, it is likely that regions of (truly critical) RK phases exist also within our $D = 5$ and $D = 7$ PEPS manifolds. Although one cannot prove that such a RK dimer liquid is realized in the $J_1-J_2$ spin-1/2 Heisenberg antiferromagnet, it is known that the critical RVB state is the ground state of a family of SU(2)-symmetric local spin-1/2 models with frustrating interactions [17, 18]. In any case, the critical dimer correlations could survive in nearby phases of some RK point over a substantial intermediate range of distances. In that case, (at least) two scenario (probably beyond our current PEPS description) may apply; First, it may well be that the dimer correlation length saturates to a (very) large value leading to a (gapped) spin liquid with $\xi_d \gg \xi_s$. A second possibility is that the system would spontaneously break translation symmetry and develops a type of (very weak) VBC ordering (dimerization, plaquette formation, ...) as suggested by large-N theories [67], series expansions [68, 69] or numerical work [21, 22, 24, 31, 32]. In fact, Lanczos ED of small clusters (see Appendix (B) for details) suggests that the tendency to realize a VBC is maximum at $J_2 \simeq 0.55$, although the VBC order parameter should be quite small, and probably very hard to detect directly. Note that a gapped spin liquid with $\xi_d \gg \xi_s$ could be alternatively seen as a melted VBC.

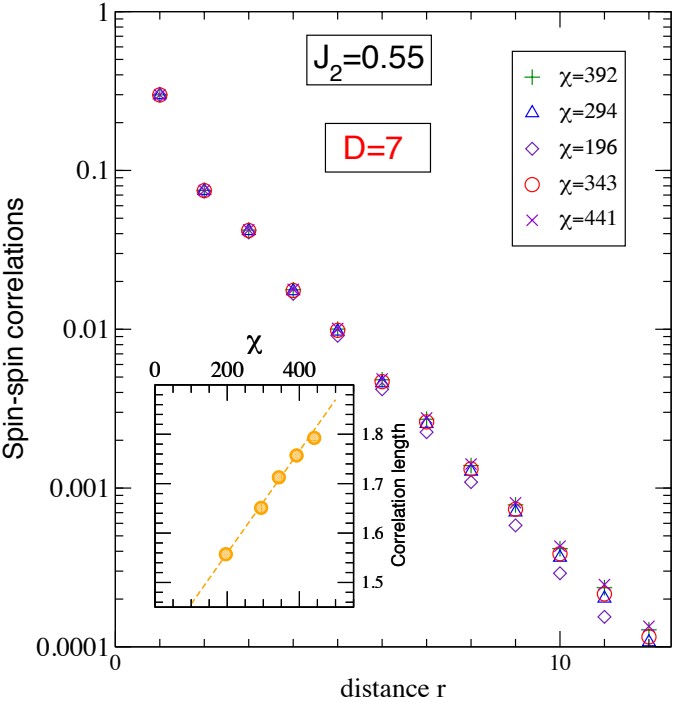

Figure 6: (a) Spin-spin correlations vs distance (on a log-log scale) for $J_2 = 0.55$, $D = 7$ and several values of the environment dimension. Inset: spin-spin correlation length vs $\chi$.

## Acknowledgments

We acknowledge useful discussions with Fabien Alet and thank Juraj Hasik and Laurens Vanderstraeten for private communications. D.P. also acknowledges inspiring conversations with Federico Becca, Paul Fendley, Zheng-Cheng Gu, Steve Kivelson, and Wen-Yuan Liu. S. C. thanks Alex Wietek for making some of the ED simulations possible using MPI technique [70].

**Funding information**    This project is supported by the TNSTRONG ANR-16-CE30-0025 and TNTOP ANR-18-CE30-0026-01 grants awarded by the French Research Council. This work was granted access to the HPC resources of CALMIP supercomputing center (under the allocations P1231 and P0677) and GENCI (Grant number A0050500225).

## A    CTMRG method

In this appendix, we provide a brief and self-contained description of the CMTRG method used to characterize the properties of the iPEPS states considered in this paper. We focus on the renormalization procedure aiming at deriving converged environment tensors (corner and edge tensors) at the thermodynamic limit that can be further used to compute states properties such as energy or correlation functions. The discussion is restricted to the case of a fully symmetric tensor (*i.e.* transforming according to the $A_1$ representation of $C_{4v}$) in the context of a translationally invariant Ansatz (the same tensor is used on every site of the square lattice). For a more general presentation, one can refer to the appendix A of reference [53].

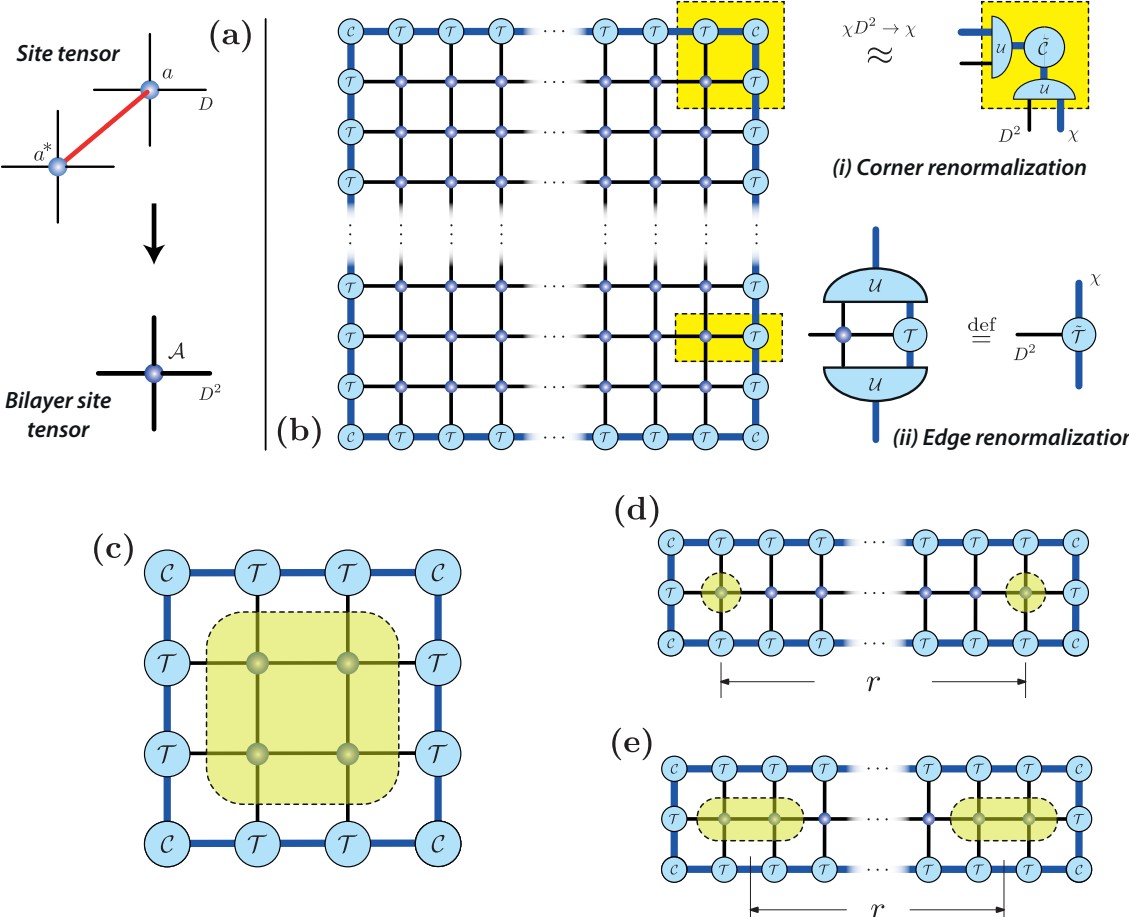

**Figure 7: (a)** The bilayer site tensor $\mathcal{A}$ is obtained by contracting the physical indices (red line) of the site tensor $a$ and its conjugate $a^*$ (note that in our case $a$ is real). **(b)** The two steps CTMRG procedure involving corner and edge tensors. The 2D lattice is contracted starting from its corners (the four corners are identical). The insertion of a site (i) is absorbed by inserting approximate isometries $\mathcal{U}$. The latter are used, in a second step, to absorb the insertion of a site on the edge tensor (ii) (see text for details).**(c-e)** Geometric setup used to compute energy **(c)**, spin-spin correlation functions **(d)** and dimer-dimer correlation functions **(e)**.

*Bilayer tensors*. In the infinite-PEPS (iPEPS) method [49], one considers a PEPS Ansatz $|\Psi\rangle$ directly in the thermodynamic limit. The PEPS is an infinite two-dimensional tensor network defined by a single site tensor, and its normalization $\langle\Psi|\Psi\rangle$ is then a bilayer tensor network which can be re-expressed as a tensor network of site rank-4 bilayer tensors (of bond dimension $D^2$). The bilayer tensor is represented in Fig. 7(a) and possesses full invariance under spin rotation and point group symmetry operations.

*Observables*. Computation of observables (like energy or correlations) also requires the bilayer tensor network which is approximately contracted over the (infinite) space surrounded a small $M$-site cluster. This approximate contraction then leads to an effective "environment" of this small region. For the energy one needs a $M = 2 \times 2 = 4$ site cluster (fitting the interaction on both NN and diagonal bonds, see Fig. 7(c)) and, for the correlations at distance $r$, a one-dimensional $r$-site segment connecting two operators at its two ends (see Fig. 7(d) and (e)).

*Renormalization procedure.* The computation of the environment is based on a Corner Transfer Matrix Renormalization Group (CTMRG) [50–53] scheme schematically represented in Fig. 7(b).

The environment involves a $\chi \times \chi$ corner transfer matrix $\mathcal{C}$ and a rank-3 boundary $\chi \times \chi \times D^2$. In practice $\chi = kD^2$ with $k$ integer. Before describing the several steps of the CTRMG algorithm let us remark that, thanks to the $A_1$ symmetry of the site tensor, several important simplifications occurs in the procedure. First of all, the four corner matrices as well as the four edge tensors are degenerate, so that a single $(\mathcal{C}, \mathcal{T})$ couple is needed. Furthermore, $\mathcal{C}$ is a real symmetric matrix. Hence it can be reduced by diagonalization (instead of a singular value decomposition) and only one isometry $\mathcal{U}$ has to be considered.

1. *Initialization step.* Corner matrix $\mathcal{C}$ and edge tensor $\mathcal{T}$ are initialized in a similar way as bilayer site tensor is constructed from the site tensor (Fig. 7(a)). In addition to the physical index, one (resp. two) virtual bonds are contracted between the two layers.

2. *Corner renormalization.* The new corner $\tilde{\mathcal{C}}$ is obtained in two steps. Starting with a $\mathcal{T}\mathcal{C}\mathcal{T}$ corner, one adds a bilayer site tensor $\mathcal{A}$ (see yellow square on Fig. 7(b)). The resulting (real) symmetric $\chi D^2 \times \chi D^2$ matrix is diagonalized and an $\chi D^2 \times \chi$ isometry $\mathcal{U}$ is constructed by keeping only (at most) the $\chi$ largest weights. Special care is taken to preserve the SU(2) spin-rotation symmetry in the truncation by keeping the SU(2) multiplet structure appearing in the corner matrix spectrum.

3. *Edge Renormalization* By adding a bilayer site tensor $\mathcal{A}$ to the edge tensor $\mathcal{T}$ and contracting with the isometry $\mathcal{U}$, the renormalized $\chi \times \chi \times D^2$ edge tensor $\tilde{\mathcal{T}}$ is obtained (see yellow rectangle on Fig. 7(b)).

Steps 2. and 3. are then repeated until a fixed point for $\mathcal{C}$ is reached. Note that the complexity of step 2. is $\chi^3 D^2 + \chi^3 D^4 + \chi^2 D^8 = k^3 D^8 + k^3 D^{10} + k^2 D^{12} \sim k^2 D^{12}$ for the untruncated corner matrix computation and $\left(\chi D^2\right)^3 = k^3 D^{12}$ for the diagonalization. The cost of step 3. is $\chi^3 D^4 + \chi^3 D^6 + \chi^2 D^8 = k^3 D^{10} + k^2 D^{12} + k^3 D^{12} \sim k^3 D^{12}$. As a result, the algorithmic complexity is $D^{12}$ for large $D$.

# B    VBC order parameters computed by ED

In order to investigate if the ground-state is a VBC that breaks lattice symmetries, we have computed the dimer-dimer correlation function:

$$C_{ijkl} = 4\left(\langle (\mathbf{S}_i \cdot \mathbf{S}_j)(\mathbf{S}_k \cdot \mathbf{S}_l)\rangle - \langle \mathbf{S}_i \cdot \mathbf{S}_j \rangle^2\right)$$

on various finite-size tori of $N$ sites. Following Ref. [22], we can then compute various structure factors, and in particular

$$S_{\text{VBC}} = \frac{1}{N_b}\sum_{k,l}\varepsilon(k,l)C_{ijkl},$$

where the summation is over $N_b$ parallel bonds $(kl)$ with respect to the reference bond $(ij)$ and $\varepsilon(k,l) = \pm 1$ depending on the sublattice, see Fig. 8(c). It can be shown that $S_{\text{VBC}}$ is finite both for a VBC with columnar or plaquette order [22].

In Fig. 8(a), we plot the behavior of $S_{\text{VBC}}$ vs $J_2$ for different finite-size clusters. In order to remove short distance data from this order parameter, we have also considered a slightly different definition

$$S_{\text{VBC}}^* = \frac{1}{N_b}\sum_{k,l}'\varepsilon(k,l)C_{ijkl},$$

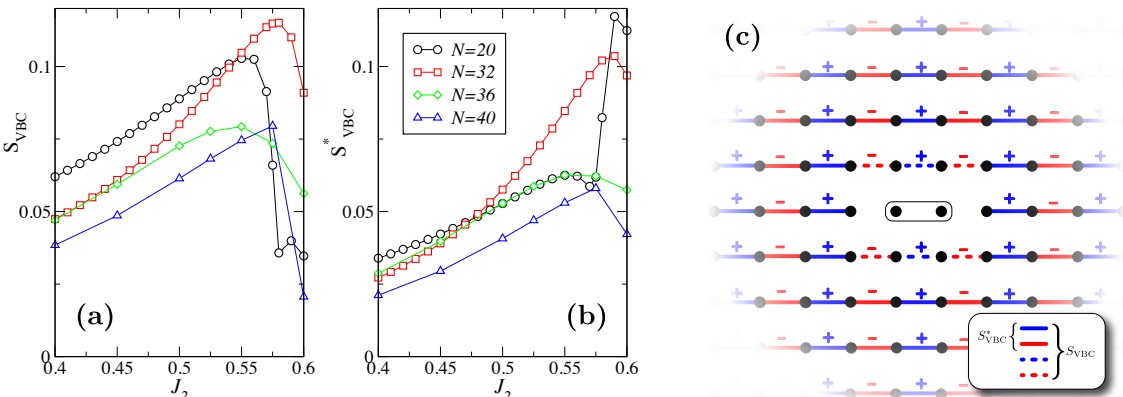

Figure 8: (a) VBC order parameter $S_{\text{VBC}}$ vs $J_2$ computed on square-shaped $\sqrt{N} \times \sqrt{N}$ tori of $N$ sites. (b) Same data for the modified VBC order parameter (see text). (c) Sign structure $\varepsilon(k, l)$ of the VBC order parameter (see text). In the modified VBC order parameter $S^*_{\text{VBC}}$, the dashed bonds are excluded in the summation.

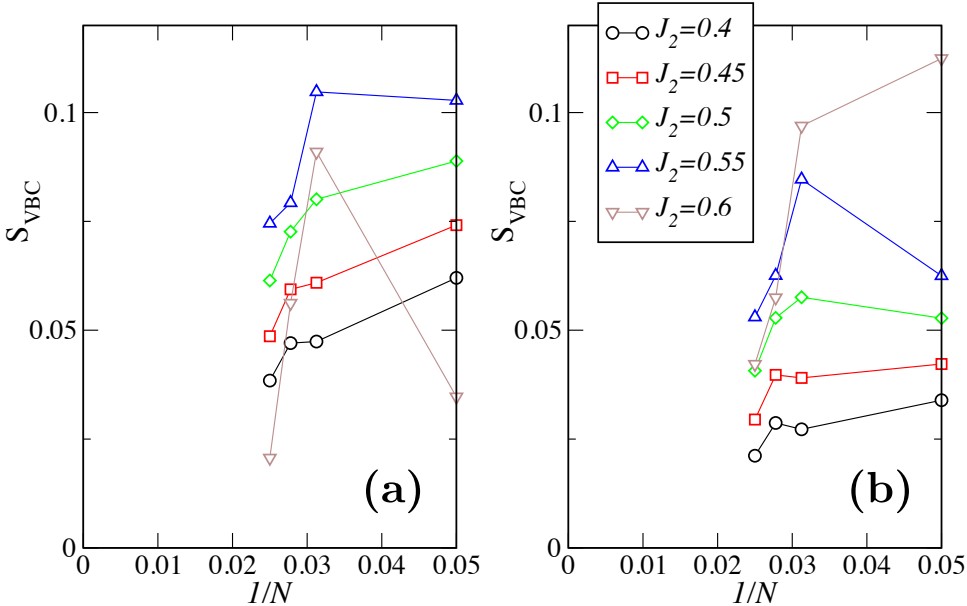

Figure 9: (a) Scaling of the VBC order parameter square $S_{\text{VBC}}$ vs $1/N$. (b) Same data for the modified VBC order parameter $S^*_{\text{VBC}}$ (see text).

where the summation does not include the nearest six bonds, see Fig. 8(b,c). Note that, compared to the ground-state energy calculations, we only computed VBC order parameter on clusters that are compatible with plaquette or columnar order, i.e. contain $(\pi, 0)$ and $(0, \pi)$ in their Brillouin zone. Quite interestingly, both VBC order parameters are maximal around $J_2 \simeq 0.55$, which is the optimal value found in DMRG [24], and then have a sudden drop beyond $J_2 = 0.6$, which is presumably of first-order character.

Finite-size scaling analysis is shown in Fig. 9 for VBC order parameters at various $J_2$ values. Reliable extrapolation is not possible, but given the data points and their curvature vs $1/N$, data are compatible with a vanishing VBC order parameter for $J_2 = 0.4$, 0.5 or 0.6, but weak long-range VBC order could be stabilized around $J_2 = 0.55$.

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
