# Peer review of "Critical colored-RVB states in the frustrated quantum Heisenberg model on the square lattice"

_SciPost Physics, doi:SciPost Phys. 7, 041 (2019)_

## Round 1 · Referee Report · Anonymous (Referee 1) · 2019-8-26

Strengths
- Paper explores an interesting and long-standing problem regarding the nature of the quantum disordered region for the spin-$1/2$ $J_1$-$J_2$ Heisenberg model on the square lattice.
- A rather careful variational wave-function analysis is presented using the iPEPS method.
- Signatures of an interesting state with short-ranged spin-spin correlations but critical dimer-dimer correlations found for $J_2/J_1 \sim 0.55$.
Weaknesses
- The technicalities related to the variational determination of the ground state using iPEPS are a bit difficult to follow.
Report
The properties of the ground state of the spin-$1/2$ $J_1$-$J_2$ Heisenberg model on the square lattice in its quantum disordered region (in the vicinity of $J_2/J_1 \sim 0.5$) has attracted considerable attention for several years now due to the interplay of strong quantum fluctuations and frustration. Despite this, the nature of the phase(s) in the quantum disordered region remains controversial.
Here, the authors consider a family of PEPS with the full space group symmetry of the lattice and the $SU(2)$ spin rotation symmetry incorporated in their calculations. Within this variational approach, they find an interesting RVB-like state with short-ranged spin-spin correlations but (almost) critical dimer-dimer correlations for $J_2/J_1 \sim 0.55$. This state is rather different from a gapless spin liquid (where the spin-spin correlations are also algebraic) obtained at $J_2/J_1=0.5$ using a similar framework.
Could the authors comment on the following? Can they detect some signatures of an enlarged symmetry (possibly $U(1)$) from the wave-function at $J_2/J_1 \sim 0.55$, e.g., by looking at columnar and plaquette like dimer correlation functions as a function of $r$? Secondly, when they compute the dimer-dimer correlations using long stripes, how do they take into account of the reduced lattice symmetry in their variational parameters?
The results presented here are definitely interesting and the numerics has been carefully done. After getting appropriate response from the authors to my questions above, I will be happy to recommend this manuscript for publication.
Here, the authors consider a family of PEPS with the full space group symmetry of the lattice and the $SU(2)$ spin rotation symmetry incorporated in their calculations. Within this variational approach, they find an interesting RVB-like state with short-ranged spin-spin correlations but (almost) critical dimer-dimer correlations for $J_2/J_1 \sim 0.55$. This state is rather different from a gapless spin liquid (where the spin-spin correlations are also algebraic) obtained at $J_2/J_1=0.5$ using a similar framework.
Could the authors comment on the following? Can they detect some signatures of an enlarged symmetry (possibly $U(1)$) from the wave-function at $J_2/J_1 \sim 0.55$, e.g., by looking at columnar and plaquette like dimer correlation functions as a function of $r$? Secondly, when they compute the dimer-dimer correlations using long stripes, how do they take into account of the reduced lattice symmetry in their variational parameters?
The results presented here are definitely interesting and the numerics has been carefully done. After getting appropriate response from the authors to my questions above, I will be happy to recommend this manuscript for publication.
Requested changes
- Few typos need to be corrected. E.g. Page 2, second paragraph->"which are specially designed to "describe" SU(2)-invariant", Page 6, first paragraph->"Hence, we have "improved" the CTMRG", Page 11, last paragraph->"(at least) two "scenarios""

---

## Round 1 · Referee Report · Anonymous (Referee 2) · 2019-8-28

Strengths
-study of an important model in the field
-state of the art results
-fresh insights for a new critical spin liquid phase
-state of the art results
-fresh insights for a new critical spin liquid phase
Weaknesses
- too technical for general audience
Report
The Authors report a numerical study of the spin-1/2 J1-J2 Heisenberg model on the square lattice, based on a family of variational, SU(2)-symmetric PEPS wavefunctions. This model is one of the first and simplest frustrated models studied, with potential spin liquid states in the region J2~J1/2. Despite many studies, the nature of the ground state(s) in this region is still unsettled.
The Authors focus at the parameter point J2=0.55J1 and offer fresh insights for a possible critical phase, different from the one at J2=0.5J1. While these results do not settle the issue fully (the Authors point out possible connections to VBC phases reported previously), this study should be interesting for people working in this field and will motivate further investigations in this very old model. I would therefore recommend the article for publication.
The paper is well written, although a bit too technical for non-specialists.
The Authors focus at the parameter point J2=0.55J1 and offer fresh insights for a possible critical phase, different from the one at J2=0.5J1. While these results do not settle the issue fully (the Authors point out possible connections to VBC phases reported previously), this study should be interesting for people working in this field and will motivate further investigations in this very old model. I would therefore recommend the article for publication.
The paper is well written, although a bit too technical for non-specialists.

---

## Round 2 · Author Response

Dear Editor,

We would like to resubmit our manuscript to SciPost. Following the referees' comments we have slightly modified the manuscript and added one ref to a relevant very recent work.

Sincerely yours,

Didier Poilblanc, on behalf of the authors

---

## Round 2 · List of Changes

1) We have rearranged a bit the presentation of the iPEPS method, to make it more accessible to non-expert (the more technical part has been left in the Appendix). In particular we have now two separate subsections, one on the real space RG method (CTMRG) used to contract the tensor network and one on the optimization procedure.

2) Following one referee's comment we have made clear that, "although we are using a strip geometry,the local tensor (and the corresponding environment tensorT) has been optimized for the fully rotationally invariant (infinite) lattice." Note that, currently, the method does not allow to measure correlations along some tilted direction with respect to the crystal axis (to capture e.g. some emergent U(1) symmetry).

3) a small number of typos have been corrected, and a reference added.

You are currently on this page

Resubmission 1907.03678v2 on 13 September 2019

---

## Editorial Decision

published